# Debiasing Deep Generative Models via Likelihood-free Importance Weighting

## Abstract

A learned generative model often gives biased statistics relative to the underlying data distribution. A standard technique to correct this bias is by importance weighting samples from the model by the likelihood ratio under the model and true distributions. When the likelihood ratio is unknown, it can be estimated by training a probabilistic classifier to distinguish samples from the two distributions. In this paper, we employ this likelihood-free importance weighting framework to correct for the bias in using state-of-the-art deep generative models.We find that this technique consistently improves standard goodness-of-fit metrics for evaluating the sample quality of state-of-the-art generative models, suggesting reduced bias. Finally, we demonstrate its utility on representative applications in a) data augmentation for classification using generative adversarial networks, and b) model-based policy evaluation using off-policy data.

## 1  Introduction

Learning probabilistic generative models of complex environments from high-dimensional observations is a long-standing challenge in machine learning. Once learned, these models are used to draw inferences and plan future actions. For example, in data augmentation, samples from a learned model are used to enrich a dataset for supervised learning (Antoniou et al., 2017). In model-based off-policy policy evaluation (henceforth MBOPE), a learned dynamics model is used to simulate and evaluate a target policy without real-world deployment Mannor et al. (2007); Thomas & Brunskill (2016), which is especially valuable for risk-sensitive applications Thomas (2015).

In recent years, deep generative models have made substantial progress in learning high-dimensional distributions and shown remarkable success across several applications in computer vision, natural language processing, and reinforcement learning (Goodfellow, 2016; Kim et al., 2018; Ho & Ermon, 2016). However, existing theoretical results (Rosenblatt, 1956; Arora et al., 2018; Zhao et al., 2018) show that learning distributions in an unbiased manner is either impossible or has prohibitive sample complexity. Consequently, the models used in practice are inherently *biased*,[1] and any downstream planning and inference based on a biased model can be misleading.

In order to address this issue, our work starts from the observation that many typical uses of generative models involve computing expectations under the generative model. For instance, in MBOPE, we seek to find the expected return of a policy under a trajectory distribution defined by this policy and learned dynamics model. A classical recipe for correcting the bias in expectations, when samples from a different distribution than the ground truth are available, is to importance weight the samples according to the likelihood ratio (Horvitz & Thompson, 1952). If the importance weights were exact, the resulting estimates are unbiased. But in practice, the likelihood ratio is unknown and needs to be estimated since the true data distribution is unknown and even the model likelihood is intractable or ill-defined for many deep generative models, e.g., variational autoencoders Kingma & Welling (2013) and generative adversarial networks Goodfellow et al. (2014).

Our proposed solution to estimate the importance weights is to train a calibrated, probabilistic classifier to distinguish samples from the true data distribution and the generative model. As has been shown in prior work, the output of such classifiers can be used to extract density ratios (Sugiyama et al., 2012). Appealingly, this estimation procedure is likelihood-free since it only requires samples

---

[1]We call a generative model biased if it produces biased statistics relative to the true data distribution.

from the two distributions. The density ratio perspective has been used previously to expand the class of learning objectives for deep generative modeling (Goodfellow et al., 2014; Nowozin et al., 2016; Mohamed & Lakshminarayanan, 2016; Grover & Ermon, 2018). While following the same estimation procedure, we use the density ratios as importance weights for bias reduction of a *pretrained* generative model to be used for downstream Monte Carlo evaluation.

Empirically, we evaluate our bias reduction framework on deep generative models of high-dimensional datasets on three main sets of experiments. First, we consider goodness-of-fit metrics for evaluating sample quality of a likelihood-based and a likelihood-free state-of-the-art model on the CIFAR-10 dataset. In particular, we experiment with PixelCNN++ (Salimans et al., 2017) and SNGAN (Miyato et al., 2018) models on three most commonly used metrics viz. Inception Scores (Salimans et al., 2016), Frechet Inception Distance (Heusel et al., 2017), and Kernel Inception Distance (Bińkowski et al., 2018). All these metrics are defined as Monte Carlo estimates from the generated samples. By importance weighting samples, we observe improvements of 23.35% and 13.48% averaged across the three metrics on the PixelCNN++ and SNGAN models respectively.

Next, we demonstrate the utility of our approach on the task of data augmentation for multi-class classification on the Omniglot dataset (Lake et al., 2015). This dataset is particularly relevant for data augmentation since it contains only 20 images per class and 1600 classes in total. We show that while naively augmenting the model with samples from a data augmentation generative adversarial network due to (Antoniou et al., 2017) is not very effective for multi-class classification, we can improve classification accuracy from 66.03% to 68.18% by importance weighting the contributions of each augmented data point.

Our final experiment demonstrates bias reduction for model-based off policy evaluation (Precup et al., 2000). A typical MBOPE approach is to first estimate a generative model of the dynamics using off-policy data and then evaluate the policy using Monte Carlo sampling (Mannor et al., 2007; Thomas & Brunskill, 2016). Again, we observe that correcting the bias of the estimated dynamics model via importance weighting leads to significantly better policy evaluations on three MuJoCo environments (Todorov et al., 2012).

## 2 Preliminaries

In this section, we discuss the necessary notation and background in deep generative modeling. Unless explicitly stated otherwise, we assume probability distributions admit absolutely continuous densities on a suitable reference measure. We use uppercase notation $X, Y, Z$ to denote random variables, lowercase notation $x, y, z$ to denote specific values in the corresponding sample spaces $\mathcal{X}, \mathcal{Y}, \mathcal{Z}$. We use boldface for multivariate random variables and their vector values.

Consider a finite dataset $D_{\text{train}}$ of instances $\mathbf{x}$ drawn i.i.d. from a fixed, but unknown distribution $p_{\text{data}}$. Given $D_{\text{train}}$, the goal of generative modeling is to learn a distribution $p_\theta$ to approximate $p_{\text{data}}$. Here, $\theta$ denotes the model parameters, e.g. weights in a neural network for deep generative models. The quality of approximation is measured via a suitable measure of discrepancy between distributions, e.g., KL divergence, Wasserstein distance, maximum mean discrepancy, moment matching (Nowozin et al., 2016; Arjovsky et al., 2017; Li et al., 2017; Ravuri et al., 2018) etc.

Broadly, there exist two main paradigms for learning a deep generative model: maximum likelihood estimation (MLE) and adversarial training (Mohamed & Lakshminarayanan, 2016). MLE fits parameters to maximize the model likelihood for the dataset $D_{\text{train}}$, when the model specifies a likelihood function, e.g. autoregressive models (Uria et al., 2016), normalizing flow models (Dinh et al., 2014), and variational autoencoder models (Kingma & Welling, 2013). Adversarial training, on the other hand, learns a generative model via a minimax game between the generative model and an auxiliary critic, where the critic distinguishes the samples in $D_{\text{train}}$ and from those generated by the model (Goodfellow et al., 2014). This method is likelihood-free since the learning objective only requires evaluating expectations w.r.t. the current model distribution during training, which can be done by drawing samples from the model.

## 3 GENERATIVE MODELS FOR MONTE CARLO EVALUATION

In this work, we are interested in use cases where the goal is to evaluate or optimize the expectations of functions under some distribution $p$ (either equal or close to the data distribution $p_{\text{data}}$). Assuming access to samples from $p$ as well some generative model $p_\theta$, one extreme is to evaluate the sample average using the samples from $p$ alone. However, this ignores the availability of $p_\theta$, to which we have a virtually unlimited access ignoring computational constraints and can arbitrarily improve the accuracy of our estimates when $p_\theta$ is close to $p$. We begin by presenting a direct motivating use case of data augmentation using deep generative models for training classifiers which generalize better. Thereafter, we discuss more generally how using generative models for evaluating Monte Carlo expectations and propose a debiasing mechanism based on importance weighting.

### 3.1 EXAMPLE USE CASE: DATA AUGMENTATION

The availability of training data is critical for learning classification and regression systems. Sufficient labeled training data may however be expensive to obtain or susceptible to noise. Data augmentation seeks to improve the performance of supervised learning systems by artificially injecting new datapoints into the training set. These new datapoints are derived from an existing labeled dataset, either by specifying manual transformations (e.g., rotations, flips for image data), or alternatively, learned via a generative model as demonstrated successfully in recent work (Ratner et al., 2017; Antoniou et al., 2017).

Consider a supervised learning task over a labeled dataset $D_{\text{cl}}$ of pairs of features and labels denoted as $(\mathbf{x}, y)$, which are assumed to be sampled independently from an underlying data distribution $p_{\text{data}}(\mathbf{x}, y)$ defined over $\mathcal{X} \times \mathcal{Y}$. Further, let $\mathcal{Y} \subseteq \mathbb{R}^k$. In order to learn a classifier $f_\phi : \mathcal{X} \to \mathbb{R}^k$, we are interested in minimizing the expectation of a loss $\ell : \mathcal{Y} \times \mathbb{R}^k \to \mathbb{R}$ over the training dataset:

$$\mathbb{E}_{p_{\text{data}}(\mathbf{x}, y)}[\ell(y, f_\phi(\mathbf{x}))] \approx \frac{1}{|D_{\text{cl}}|} \sum_{(\mathbf{x}, y) \sim D_{\text{cl}}} \ell(y, f_\phi(\mathbf{x})). \tag{1}$$

For example, $\ell$ could be specified as the per example cross-entropy loss. Optimizing the above objective w.r.t $\phi$ hence requires reliably estimating the function $f_\phi \in \mathcal{F}$ that best fits the data distribution.

A generative model for the task of data augmentation learns a joint distribution $p_\theta(\mathbf{x}, y)$. Several algorithmic variants exist for learning the model's joint distribution and we defer the specifics of prior work to the experiments section. Once the generative model is learned, it can be used to optimize the expected classification loss in Eq. 1 under a mixture distribution of empirical data distributions and generative model distributions given as:

$$p_{\text{mix}}(\mathbf{x}, y) = m p_{\text{data}}(\mathbf{x}, y) + (1 - m) p_\theta(\mathbf{x}, y) \tag{2}$$

for a suitable choice of the mixture weights $m \in [0, 1]$. Data augmentation is a standard routine in most supervised learning problems, and recent work has successfully applied it in low labelled data regimes (Wong et al., 2016; Antoniou et al., 2017). Notice that while the eventual task here is optimization, reliably evaluating the expected loss of a candidate parameter $\phi$ is an important ingredient and we focus on this basic question first, before leveraging the solution for data augmentation and other use cases. Also observe that the distribution $p$ under which we seek expectations is same as $p_{\text{data}}$ here, and we rely on the generalization ability of $p_\theta$ to generate transformations of an instance in the dataset which are not explicitly present, but plausibly observed in other, similar instances (Zhao et al., 2018).

### 3.2 DEBIASING USING IMPORTANCE WEIGHTS

Whenever the distribution $p$ under which we seek expectations differs from $p_\theta$, model-based estimates exhibit bias. In this section, we start out by formalizing bias for Monte Carlo expectations and subsequently propose a bias reduction strategy based on likelihood-free importance weighting. The notation used in this section follows Section 2.

We are interested in evaluating expectations of a class of functions of interest $f \in \mathcal{F}$ w.r.t. the distribution $p$. For any given $f : \mathcal{X} \to \mathbb{R}$, we get:

$$\mathbb{E}_{\mathbf{x} \sim p}[f(\mathbf{x})] = \int p(\mathbf{x}) f(\mathbf{x}) \mathrm{d}\mathbf{x}. \tag{3}$$

Given access to samples from a generative model $p_\theta$, if we knew the densities for both $p$ and $p_\theta$, then a classical scheme to evaluate expectations under $p$ using samples from $p_\theta$ is to use importance sampling (Horvitz & Thompson, 1952). We reweight each sample from $p_\theta$ according to its likelihood ratio under $p$ and $p_\theta$ and compute a weighted average of the function $f$ over these samples. Formally, we have:

$$\mathbb{E}_{\mathbf{x} \sim p}[f(\mathbf{x})] = \mathbb{E}_{\mathbf{x} \sim p_\theta}\left[ \frac{p(\mathbf{x})}{p_\theta(\mathbf{x})} f(\mathbf{x}) \right] \qquad \approx \frac{1}{T} \sum_{i=1}^{T} w(\mathbf{x}_i) f(\mathbf{x}_i) \tag{4}$$

where $w(\mathbf{x}_i) := {p(\mathbf{x}_i)}/{p_\theta(\mathbf{x}_i)}$ is the importance weight for $\mathbf{x}_i \sim p_\theta$. The validity of this procedure is subject to the use of a proposal $p_\theta(\mathbf{x})$ that for all $\mathbf{x} \in \mathcal{X}$ where $p_\theta(\mathbf{x}) = 0$, we also have $f(\mathbf{x}) p(\mathbf{x}) = 0$.[2]

However, in order to apply this technique to reduce the bias of a generative sampler $p_\theta$ w.r.t. $p$, we require knowledge of the importance weights $w(\mathbf{x})$ for any $\mathbf{x} \sim p_\theta$. However, we typically only have a sampling access to $p$. For instance, in the data augmentation example above, where $p = p_{\text{data}}$, the unknown distribution used to learn $p_\theta$. Similarly in MBOPE, $p$ involves the unknown dynamics of the environment which we can only observe samples from. Hence we need a scheme to learn the weights $w(\mathbf{x})$, using samples from $p$ and $p_\theta$, which is the problem we tackle next.

Consider two sets of samples from the distributions $p$ and $p_\theta$ respectively. Without loss of generality, assign the positive label $y = 1$ to samples from $p$ and negative label $y = -1$ to samples from $p_\theta$. A probabilistic, binary classifier $c : \mathcal{X} \to [0, 1]$ assigns a probability that a sample $\mathbf{x}$ belongs to the positive class $y = 1$. As shown in prior work, such a classifier can be used to extract density ratios (Friedman et al., 2001). We restate the result in the proposition below.

**Proposition 1.** *If a probabilistic classifier $c : \mathcal{X} \to [0, 1]$ trained to classify data from $p$ and $p_\theta$ is Bayes optimal, then the ratio of densities assigned to any point $\mathbf{x}$ is given as:*

$$\frac{p(\mathbf{x})}{p_\theta(\mathbf{x})} = \gamma \frac{c(\mathbf{x})}{1 - c(\mathbf{x})} \tag{5}$$

*where $\gamma = \frac{p(y=-1)}{p(y=1)}$.*

For the rest of the work, we assume for the purpose of brevity that a data point is equally likely to be classified as positive or negative, and hence $\gamma = 1$.[3]

### 3.3 Practical considerations

In practice, we do not have access to a Bayes optimal classifier and hence, the estimated importance weights will not be exact and consequently, we can hope to reduce the bias as opposed to eliminating it entirely. Hence, our proposed bias reduced estimator w.r.t. $p_{\text{data}}$ is given as:

$$\mathbb{E}_{\mathbf{x} \sim p}[f(\mathbf{x})] \approx \frac{1}{T} \sum_{i=1}^{T} \hat{w}(\mathbf{x}_i) f(\mathbf{x}_i) \tag{6}$$

where $\hat{w}(\mathbf{x}_i) = {c(\mathbf{x}_i)}/{(1 - c(\mathbf{x}_i))}$ is the importance weight for $\mathbf{x}_i \sim p_\theta$ estimated via a probabilistic binary classifier $c(\mathbf{x})$.

---

[2]A stronger sufficient, but not necessary condition that is independent of $f$, states that the proposal $p_\theta$ is valid if it has a support larger than $p$, i.e., for all $\mathbf{x} \in \mathcal{X}$, $p_\theta(\mathbf{x}) = 0$ implies $p(\mathbf{x}) = 0$.

[3]This can be enforced empirically by training a classifier on an equal number of positive and negative examples.

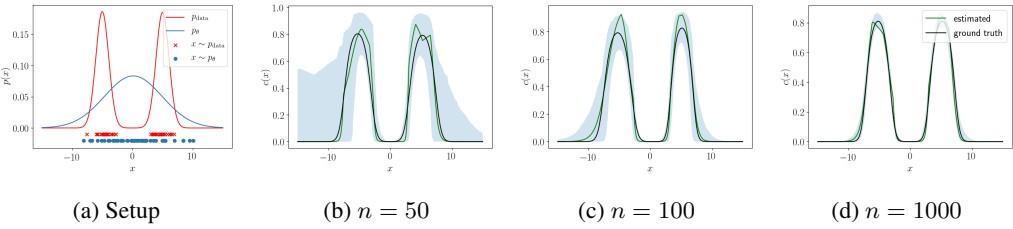

| (a) Setup | (b) $n = 50$ | (c) $n = 100$ | (d) $n = 1000$ |

Figure 1: Importance Weight Estimation using Probabilistic Classifiers. (a) A univariate Gaussian (blue) is fit to samples from a mixture of two Gaussians (red). (b-d) Estimated class probabilities (with 95% confidence intervals based on 1000 bootstraps) for varying number of points $n$, where $n$ is the number of points used for training the generative model and multilayer perceptron.

**Self-normalization.**    If the importance weights across the sample set are too small, one useful trick from the importance sampling literature that works well in practice is to normalize the importance weights across a batch.

The self-normalized importance weighted estimator for Monte Carlo evaluation is given as:

$$\mathbb{E}_{\mathbf{x}\sim p}[f(\mathbf{x})] \approx \sum_{i=1}^{T} \frac{\hat{w}(\mathbf{x}_i)}{\sum_{j=1}^{T} \hat{w}(\mathbf{x}_j)} f(\mathbf{x}_i). \tag{7}$$

where $\hat{w}(\mathbf{x}_i)$ is the importance weight for $\mathbf{x}_i \sim p_\theta$.

**Deriving confidence intervals.**    To derive confidence intervals around the estimated importance weights or $c(\mathbf{x})$, we propose to use a combination of empirical and parametric bootstraps. Bootstrap is a widely-used tool in statistics for deriving confidence intervals by fitting ensembles of models on resampled data points. If the dataset is finite e.g., $D_{\text{train}}$, then the bootstrapped dataset is obtained via random sampling *with replacement* and confidence intervals are estimated via the *empirical bootstrap*. For a parameteric model generating the dataset e.g., $p_\theta$, a fresh bootstrapped dataset is resampled from the model and confidence intervals are estimated via the *parametric bootstrap*. See Efron & Tibshirani (1994) for a detailed review. In training a binary classifier as described above, we can estimate the confidence intervals by retraining the classifier on a fresh sample of points from $p_\theta$ and a resampling of the training dataset $D_{\text{train}}$ (with replacement).

**Synthetic experiment.**    We visually illustrate the efficacy of classifiers for estimating importance weighting in a toy experiment. The setup for this experiment is illustrated in Figure 1a. We are given a finite set of samples drawn from a mixture of two Gaussians (red). The model family is a unimodal Gaussian with two parameters, illustrating mismatch due to a parametric model. The mean and variance of the model are estimated by the empirical means and variances of the observed data. Using estimated model parameters, we then draw samples from the model (blue).

In Figure 1b, we show the probability assigned by a binary classifier to a point to be from true data distribution. Here, the classifier is a multi-layer perceptron with a single hidden layer of 100 units and has been trained by gradient-based methods on a dataset of 50 samples drawn from the generative model and data distribution each. The density ratios are estimated based on the probabilities assigned by the classifier. The classifier is not Bayes optimal, which can be seen by the gaps between the optimal probabilities curve (black) and the estimated class probability curve (green). However, as we increase the number of real and generated examples $n$ in Figure 1c,d from $n = 50$ to $n = 100$ and $n = 1000$, the classifier approaches optimality. Furthermore, even its uncertainty shrinks with increasing data, as expected. In summary, this experiment demonstrates model mismatch in a generative model as the root cause of bias and how a binary classifier can mitigate this bias.

## 4    EXPERIMENTS

Our experiments are designed to demonstrate two key takeaways: (a) likelihood-free importance weighting can reduce bias of deep generative models on standard goodness-of-fit metric evaluations,

Table 1: Goodness-of-fit evaluation on CIFAR-10 dataset for PixelCNN++ and SNGAN. Standard errors computed over 10 runs. **Higher IS is better. Lower FID and KID scores are better.**

| Model | Evaluation | IS ($\uparrow$) | FID ($\downarrow$) | KID ($\downarrow$) |
|---|---|---|---|---|
| - | Reference | $11.09 \pm 0.1263$ | $5.20 \pm 0.0533$ | $0.008 \pm 0.0004$ |
| PixelCNN++ | Default | $5.16 \pm 0.0117$ | $58.70 \pm 0.0506$ | $0.196 \pm 0.0001$ |
| | IW | $\mathbf{6.68} \pm 0.0773$ | $\mathbf{55.83} \pm 0.9695$ | $\mathbf{0.126} \pm 0.0009$ |
| SNGAN | Default | $8.33 \pm 0.0280$ | $20.40 \pm 0.0747$ | $0.094 \pm 0.0002$ |
| | IW | $\mathbf{8.57} \pm 0.0325$ | $\mathbf{17.29} \pm 0.0698$ | $\mathbf{0.073} \pm 0.0004$ |

(b) debiasing improves standard approaches to using deep generative models for data augmentation and model-based off-policy policy evaluation. In all our experiments, the binary classifier used to estimate the importance weights was a deep neural network trained based on the binary cross-entropy loss, and we found it useful to normalize the estimated importance weights. Further, we ensured that *the classifiers used were well-calibrated*. Hyperparameter details for the experiments beyond those mentioned here are deferred to the appendices.

## 4.1 GOODNESS-OF-FIT TESTING

In the first set of experiments, we highlight the benefits of importance weighting for debiasing transfer over to deep generative models trained on high-dimensional data distributions. Since our debiasing strategy is agnostic to the choice of the downstream function for which we wish to perform Monte Carlo averaging, we experimented with three popularly used metrics for evaluating generative models viz. Inception Scores (IS) (Salimans et al., 2016), Frechet Inception Distance (FID) (Heusel et al., 2017), and Kernel Inception Distance (KID) (Bińkowski et al., 2018).

For a semantic evaluation of difference in sample quality, this test is performed in the feature space of a pretrained classifier, such as the prefinal activations of the Inception Net (Szegedy et al., 2016). All these scores can be formally expressed as empirical expectations with respect to the model. For example, the Inception score for a generative model $p_\theta$ given a classifier $d(\cdot)$ can be expressed as:

$$\text{IS} = \exp(\mathbb{E}_{\mathbf{x} \sim p_\theta}[\text{KL}(d(y|\mathbf{x}), d(y))]).$$

The FID score is another metric which unlike the Inception score also takes into account real data from $p_{\text{data}}$. Mathematically, the FID between sets $S$ and $R$ sampled from distributions $p_\theta$ and $p_{\text{data}}$ respectively, is defined as:

$$\text{FID}(S, R) = \|\mu_S - \mu_R\|_2^2 + \text{Tr}(\Sigma_S + \Sigma_R - 2\sqrt{\Sigma_S \Sigma_R})$$

where $(\mu_S, \Sigma_S)$ and $(\mu_R, \Sigma_R)$ are the empirical means and covariances computed based on $S$ and $R$ respectively. Here, $S$ and $R$ are sets of datapoints from $p_\theta$ and $p_{\text{data}}$. In a similar vein, KID compares statistics between samples in a feature space defined via a combination of kernels and a pretrained classifier.

For all these metrics, we can simulate the population level unbiased case as a "reference score" wherein we artificially set both the real and generated sets of samples used for evaluation as finite, disjoint sets derived from $p_{\text{data}}$. This gives a sense of the limitations of these tests due to finite sample effects. The closer the scores for a model are to the reference score, the lesser is the bias of this model w.r.t. the particular goodness-of-fit metric under consideration.

We evaluate the three metrics for two representative state-of-the-art models trained on the CIFAR-10 dataset viz. an autoregressive model PixelCNN++ (Salimans et al., 2017) learned via maximum likelihood estimation and a latent variable model SNGAN (Miyato et al., 2018) learned via adversarial training. In order to evaluate each of these metrics, we draw 10,000 samples from the model. In Table 1, we report the metrics with and without the bias correction due to likelihood-free importance weighting (IW). Our results show that importance weighted samples are much closer to the ones observed during training for evaluating goodness-of-fit statistics, suggesting the utility of our approach even for other downstream use cases which we discuss next.

Table 2: Multi-class classification accuracy for data augmentation on the Omniglot dataset. Standard errors computed over 5 runs.

| Training Dataset | Accuracy |
|---|---|
| $D_{\mathrm{cl}}$ | $0.6603 \pm 0.0012$ |
| $D_{\mathrm{g}}$ | $0.4431 \pm 0.0054$ |
| $D_{\mathrm{g}} + \mathrm{IW}$ | $0.4481 \pm 0.0056$ |
| $D_{\mathrm{cl}} + D_{\mathrm{g}}$ | $0.6600 \pm 0.0040$ |
| $D_{\mathrm{cl}} + D_{\mathrm{g}} + \mathrm{IW}$ | $\mathbf{0.6818} \pm 0.0022$ |

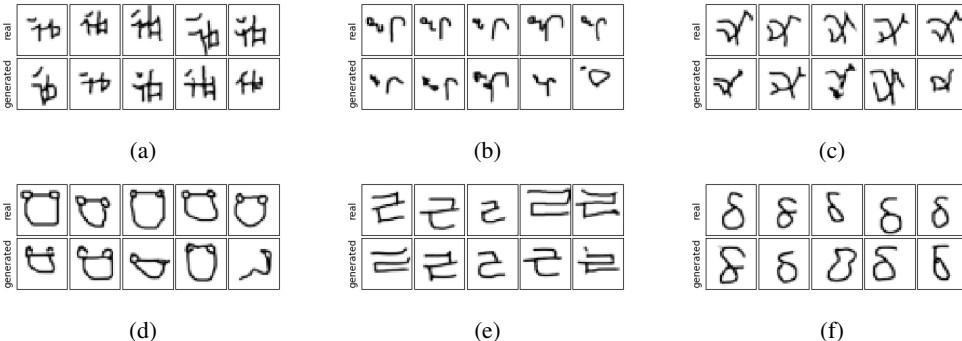

Figure 2: Qualitative evaluation of importance weighting for data augmentation. (a-f) Top row shows held-out data samples from a specific class in Omniglot. Bottom row shows generated samples from the same class *ranked in decreasing order* of importance weights.

## 4.2 DATA AUGMENTATION

A few different generative models that have demonstrated success in this task and could potentially benefit from the current work. We restrict our experiments to the use of Data Augmentation Generative Adversarial Networks (DAGAN) (Antoniou et al., 2017) as our generative model. Here, we also note that while the aforementioned work was motivated by and evaluated for the task of meta-learning, it can also be applied for multi-class classification scenarios, which is the setting we consider here. A self-contained description of DAGAN is provided in the Appendix.

We trained a DAGAN on the Omniglot dataset of handwritten characters (Lake et al., 2015). The dataset is particularly relevant because it contains 1600+ classes but only 20 examples from each class and hence, could potentially benefit from augmented data. We used the first 1200 classes for our experiments to be consistent with prior uses of this dataset, and split the 20 examples for each class into 15 training, 3 validation, and 2 test examples. All models (DAGAN, binary classifier for importance weighting, final multi-class classifier) were trained on the training examples alone with hyperparameter choices made based on the validation set.

Once the model has been trained, it can be used for data augmentation in many ways. In particular, we consider ablation baselines that use various combinations of the real training data $D_{\mathrm{cl}}$ and generated data $D_{\mathrm{g}}$ for training a downstream classifier. When the generated data $D_{\mathrm{g}}$ is used, we can either use the data directly with uniform weighting for all training points, or choose to importance weight (IW) the contributions of the individual training points to the overall loss. The results are shown in Table 2. While generated data ($D_{\mathrm{g}}$) alone cannot be used to obtain competitive performance relative to the real data ($D_{\mathrm{cl}}$) on this task as expected, the bias it introduces for evaluation and subsequent optimization overshadows even the naive data augmentation ($D_{\mathrm{cl}} + D_{\mathrm{g}}$). In contrast, we can obtain significant improvements by importance weighting the generated points ($D_{\mathrm{cl}} + D_{\mathrm{g}} + IW$).

Qualitatively, we can observe the effect of importance weighting in Figure 2. Here, we show true and generated samples for 6 randomly choosen classes (a-f) in the Omniglot dataset. The generated samples are further ranked in decreasing order of the importance weights. While there is no way to formally test the validity of such rankings, it must also be noted that this criteria can also prefer points

which have high density under $p_{\text{data}}$ but are unlikely under $p_\theta$ since we are looking at ratios. Visual inspection suggests that the classifier is able to appropriately downweight poorer samples, as shown in Figure 2 (a, b, c, d, bottom right). There are also failure modes, such as the lowest ranked generated images in Figure 2 (e, f, bottom right) where the classifier weights reasonable generated samples poorly relative to others. This could be due to particular artifacts such as a tiny disconnected blurry speck in Figure 2 (e, bottom right) which are potentially more revealing to a classifier distinguishing real and generated data.

### 4.3 MODEL-BASED OFF-POLICY POLICY EVALUATION

So far, we have seen the benefits of our debiasing framework in cases where the generative model was trained on data from the same distribution as the one we wish to use for downstream unbiased Monte Carlo evaluation. We can extend the same principle to more involved settings when the generative model is a building block for specifying the full data generation process, e.g, trajectory data generated via a probabilistic dynamics model along with an agent policy.

In particular, we consider the setting of off-policy policy evaluation (OPE), where the goal is to evaluate policies using experiences collected from a different policy. Formally, let $(\mathcal{S}, \mathcal{A}, r, P, \eta, T)$ denote an (undiscounted) Markov decision process with state space $\mathcal{S}$, action space $\mathcal{A}$, reward function $r$, transition $P$, initial state distribution $\eta$ and horizon $T$. Assume $\pi_e : \mathcal{S} \times \mathcal{A} \rightarrow [0, 1]$ is a known policy that we wish to evaluate. The probability of generating a certain trajectory $\tau = \{\mathbf{s}_0, \mathbf{a}_0, \mathbf{s}_1, \mathbf{a}_1, ..., \mathbf{s}_T, \mathbf{a}_T\}$ of length $T$ with policy $\pi_e$ and transition $P$ is given as:

$$p^\star(\tau) = \eta(\mathbf{s}_0) \prod_{t=0}^{T-1} \pi_e(\mathbf{a}_t|\mathbf{s}_t) P(\mathbf{s}_{t+1}|\mathbf{s}_t, \mathbf{a}_t). \tag{8}$$

The return on a trajectory $R(\tau)$ is the sum of the rewards across the state, action pairs in $\tau$: $R(\tau) = \sum_{t=1}^{T} r(\mathbf{s}_t, a_t)$, where we assume a *known reward function* $r$. Denoting the distribution over trajectories induced by $\pi_e$ as $p^*$, we are interested in the value of a policy defined as:

$$v(\pi_e) = \mathbb{E}_{\tau \sim p^*(\tau)} [R(\tau)]. \tag{9}$$

Evaluating $\pi_e$ using Eq. 9 requires the (unknown) transition dynamics $P$. The dynamics model is a conditional generative model of the next states $\mathbf{s}_{t+1}$ conditioned on the previous state-action pair $(\mathbf{s}_t, \mathbf{a}_t)$. If we have access to historical logged data $D_\tau$ of trajectories $\tau = \{\mathbf{s}_0, \mathbf{a}_0, \mathbf{s}_1, \mathbf{a}_1, \dots, \}$ from some behavioral policy $\pi_b : \mathcal{S} \times \mathcal{A} \rightarrow [0, 1]$, then we can use this off-policy data to train a dynamics model $P_\theta(\mathbf{s}_{t+1}|\mathbf{s}_t, \mathbf{a}_t)$. The policy $\pi_e$ can then be evaluated under this learned dynamics model.

$$\tilde{v}(\pi_e) = \mathbb{E}_{\tau \sim \tilde{p}(\tau)}[R(\tau)],$$

where $\tilde{p}$ uses $P_\theta$ instead of the true dynamics in Eq. 8. However, the trajectories sampled with $P_\theta$ could significantly deviate from samples from $P$ due to compounding errors (Ross & Bagnell, 2010). In order to correct for this bias, we can use likelihood-free importance weighting. The binary classifier $c(\mathbf{s}_t, \mathbf{a}_t, \mathbf{s}_{t+1})$ for estimating the importance weights in this case distinguishes between triples of true and generated transitions. For any true triple $(\mathbf{s}_t, \mathbf{a}_t, \mathbf{s}_{t+1})$ extracted from the off-policy data, the corresponding generated triple $(\mathbf{s}_t, \mathbf{a}_t, \hat{\mathbf{s}}_{t+1})$ only differs in the final transition state, i.e., $\hat{\mathbf{s}}_{t+1} \sim P_\theta(\hat{\mathbf{s}}_{t+1}|\mathbf{s}_t, \mathbf{a}_t)$. Such a classifier allows us to obtain the importance weights $\hat{w}(\mathbf{s}_t, \mathbf{a}_t, \hat{\mathbf{s}}_{t+1})$ for every predicted state transition $(\mathbf{s}_t, \mathbf{a}_t, \hat{\mathbf{s}}_{t+1})$.

The bias reduced estimator for OPE can then be derived as:

$$v(\pi_e) = \mathbb{E}_{\tau \sim \tilde{p}(\tau)} \left[ \frac{p^\star(\tau)}{\tilde{p}(\tau)} R(\tau) \right]. \tag{10}$$

The importance weights for the trajectory $\tau$ can be derived from the importance weights of the individual transitions:

$$\frac{p^\star(\tau)}{\tilde{p}(\tau)} = \frac{\prod_{t=0}^{T-1} P(\mathbf{s}_{t+1}|\mathbf{s}_t, \mathbf{a}_t)}{\prod_{t=0}^{T-1} P_\theta(\mathbf{s}_{t+1}|\mathbf{s}_t, \mathbf{a}_t)} = \prod_{t=0}^{T-1} \frac{P(\mathbf{s}_{t+1}|\mathbf{s}_t, \mathbf{a}_t)}{P_\theta(\mathbf{s}_{t+1}|\mathbf{s}_t, \mathbf{a}_t)}$$

$$\approx \prod_{t=0}^{T-1} \frac{c(\mathbf{s}_t, \mathbf{a}_t, \mathbf{s}_{t+1})}{1 - c(\mathbf{s}_t, \mathbf{a}_t, \mathbf{s}_{t+1})} = \prod_{t=0}^{T-1} \hat{w}(\mathbf{s}_t, \mathbf{a}_t, \hat{\mathbf{s}}_{t+1}). \tag{11}$$

Table 3: Off-policy policy evaluation on MuJoCo tasks. Standard error is over 100 trajectories used for Monte Carlo estimation.

| Environment | $v(\pi_e)$ (Ground truth) | $\tilde{v}(\pi_e)$ | $\hat{v}(\pi_e)$ (Ours) | $\hat{v}_{80}(\pi_e)$ (Ours) |
|---|---|---|---|---|
| Swimmer | $36.7 \pm 0.1$ | $16.5 \pm 16.5$ | $\mathbf{38.9} \pm 23.3$ | $57.6 \pm 34.9$ |
| HalfCheetah | $185.0 \pm 2.56$ | $129.7 \pm 1.24$ | $149.6 \pm 49.7$ | $\mathbf{152.0} \pm 78.5$ |
| HumanoidStandup | $14170 \pm 53$ | $8504 \pm 74$ | $9515 \pm 4890$ | $\mathbf{10049} \pm 7335$ |

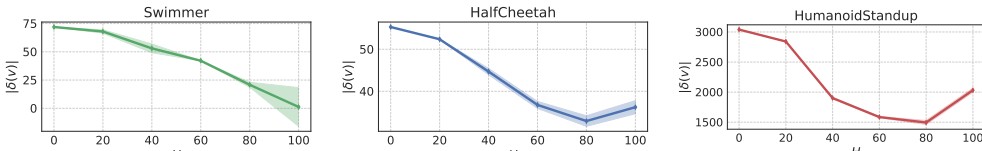

Figure 3: Estimation error $\delta(v) = |v(\pi_e) - \hat{v}_H(\pi_e)|$ under different values of $H$ (minimum 0, maximum 100). Shaded area denotes standard error over classifiers trained with different random seeds.

Our final importance weighted (IW) estimator is given as:

$$\hat{v}(\pi_e) = \mathbb{E}_{\tau \sim \tilde{p}(\tau)} \left[ \prod_{t=0}^{T-1} \hat{w}(\mathbf{s}_t, \mathbf{a}_t, \hat{\mathbf{s}}_{t+1}) \cdot R(\tau) \right]. \tag{12}$$

We consider three continuous control tasks in the MuJoCo simulator (Todorov et al., 2012) from OpenAI gym (Brockman et al., 2016) (in increasing number of state dimensions): Swimmer, HalfCheetah and HumanoidStandup. These environments have high dimensional states (e.g. HumanoidStandup has 376 dimensions), which makes learning a reliable dynamics model for OPE challenging. We train behavioral and evaluation policies using Proximal Policy Optimization (Schulman et al., 2017) with different hyperparameters for the two policies, and collect dataset from the behavior policy to train a ensemble neural network dynamics model using three fully-connected layers and swish activation functions (Ramachandran et al., 2018). We the use the trained dynamics model to evaluate $\tilde{v}(\pi_e)$ and its IW version $\hat{v}(\pi_e)$, and compare them with the ground truth returns $v(\pi_e)$. Each estimation is averaged over a set of 100 trajectories with horizon $T = 100$. Specifically, for $\hat{v}(\pi_e)$, we also average the estimation over 5 classifier instances trained with different random seeds. We further consider performing IW over only the first $H$ steps, and use uniform weights for the remainder, which we denote as $\hat{v}_H(\pi_e)$. This allow us to interpolate between $\tilde{v}(\pi_e) \equiv \hat{v}_0(\pi_e)$ and $\hat{v}(\pi_e) \equiv \hat{v}_T(\pi_e)$. Finally, as in the other experiments, we used the self-normalized variant (Eq. 7) of the importance weighted estimator in Eq. 12.

Table 3 compares the policy evaluations under different environments. These results show that the reward estimations with the trained dynamics model differ from the ground truth by a large margin. By importance weighting the trajectories, we are able to obtain much more accurate evaluations of the policy. As expected, we also see that while IW leads to higher returns on average, the imbalance in trajectory importance weights due to the multiplicative importance weights of the state-action pairs can lead to higher variance in the importance weighted returns. In Figure 3, we demonstrate that policy evaluation becomes more accurate as more timesteps are used for IW evaluations, until around $80 - 100$ timesteps and thus empirically validates the benefits of importance weighting using a classifier. Given that our estimates have a large variance, but generally include the true policy value within the uncertainty interval, it would be worthwhile to compose our approach with other variance reduction techniques such as (weighted) doubly robust estimation in future work, as well as incorporate these estimates within a framework such as MAGIC to further blend with model-free OPE (Thomas & Brunskill, 2016).

**Overall.** Across all our experiments, we observe that importance weighting the generated samples leads to uniformly better results, whether in terms of the quality of samples, or their utility in downstream tasks. Since the technique is a black-box wrapper around any generative model, we expect this to benefit a diverse set of tasks in follow-up works.

However, there is also some caution to be excercised with these techniques as evident from the results of Table 1. Note that in this table, the confidence interval (computed using the reported standard errors) around our model score after importance weighting still does not contain the reference scores using a hold-out sample from the true model. This would not have been the case if our debiased estimator was completely unbiased and this observation reiterates our earlier claim that likelihood-free importance weighting is reducing bias, as opposed to completely eliminating it. Indeed, when such a mismatch is observed, it is a good diagnostic to either construct more powerful classifiers to better approximate the Bayes optimum, or find additional data from the true distribution in case the generative model fails the full support assumption.

## 5 RELATED WORK AND DISCUSSION

Density ratios enjoy widespread use across machine learning e.g., covariate shifts (Sugiyama et al., 2012) and their estimation of via binary classifiers is frequently used for defining learning objectives for generative models. See Mohamed & Lakshminarayanan (2016) for an excellent review. In particular, such classifiers have been used to define learning fameworks such as generative adversarial networks (Goodfellow et al., 2014; Nowozin et al., 2016), likelihood-free Approximate Bayesian Computation (ABC) (Gutmann & Hyvärinen, 2012) and earlier work in unsupervised-as-supervised learning (Friedman et al., 2001) and noise contrastive estimation (Gutmann & Hyvärinen, 2012) among others. The key difference is that these works are explicitly interested in *learning* the parameters of a generative model. In contrast, we use the binary classifier for estimating importance weights to correct for the bias of any *fixed* generative model.

Classifiers have also been used for defining two-sample tests (Gretton et al., 2007; Bowman et al., 2015; Lopez-Paz & Oquab, 2016; Danihelka et al., 2017; Rosca et al., 2017; Im et al., 2018; Gulrajani et al., 2018). These are not particularly restricted to probabilistic classifiers and the goal here is to evaluate sample quality by goodness-of-fit tests, e.g., FID, KID etc. In our setting, the underlying functions are not limited to goodness-of-fit testing but could apply to arbitrary functions such as a classification loss or the value function of a policy. Finally, as shown in the experiments, our approach can also be used for a bias sensitive evaluation of the above metrics.

Closely related to the above use case are recent works by Tao et al. (2018), Azadi et al. (2018) and Turner et al. (2018) that use rejection sampling and MCMC to explicitly reject or transform the generated samples. These methods require extra computation beyond training a classifier, in rejecting the samples or running Markov chains to convergence, unlike the importance weighting strategy proposed in this work. Moreover, principled rejection sampling requires an upper bound on the density ratio that holds for all data points, which is typically infeasible to obtain.

## 6 CONCLUSION

In this work, we identified bias w.r.t. a target data distribution as a fundamental challenge restricting the use of deep generative models as proposal distributions for Monte Carlo evaluation. We proposed a debiasing framework based on importance sampling. The importance weights are learned in a likelihood-free fashion via a binary classifier distinguishing samples from the target distribution and the learned model. Empirically, we find the bias correction to be useful across a surprising variety of tasks including goodness-of-fit sample quality tests and the motivating use cases of data augmentation and model-based off-policy policy evaluation.

One interesting direction for future work is to design objectives for generative models which explicitly take into account the class of functions for which we wish to use these models as Monte Carlo proposals during learning itself. Furthermore, the ability to characterize the bias of a deep generative model is an important step towards using these models in risk-sensitive applications with high uncertainty (Gal & Ghahramani, 2016; Lakshminarayanan et al., 2017). Applying our debiasing strategy in conjunction with recent methods proposed for robust downstream inference tasks using deep generative models, such as anomaly detection, is another direction for future work (Nalisnick et al., 2018; Choi & Jang, 2018).

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

APPENDICES

## A EXPERIMENTAL DETAILS

Our codebase was implemented using the Pytorch library (Paszke et al., 2017). For the case of model-based off-policy policy evaluation experiments, we used Tensorflow (Abadi et al., 2016) and OpenAI baselines (Dhariwal et al., 2017).

### A.1 GOODNESS-OF-FIT TESTING

We used the open-sourced model implementations of PixelCNN++ (Salimans et al., 2016) and SNGAN (Miyato et al., 2018). Following the observation by Lopez-Paz & Oquab (2016), we found that training a binary classifier on top of the feature space of any pretrained image classifier was useful for removing the low-level artifacts in the generated images in classifying an image as real or fake. Learning was done using the Adam optimizer with the default hyperparameters with a learning rate of $0.001$ and a batch size of $64$. We observed relatively fast convergence for training the binary classifier (in less than 20 epochs) on both PixelCNN++ and SNGAN generated data and the validation set accuracy across the first 20 epochs was used for selecting the best checkpoint.

### A.2 DATA AUGMENTATION

A DAGAN learns to augment data by training a conditional generative model $G_\theta : \mathcal{X} \times \mathcal{Z} \to \mathcal{X}$ based on a training dataset $D_{\mathrm{cl}}$. The generative model is learned via a minimax game with a critic. For any conditioning datapoint $\mathbf{x}_i \in D_{\mathrm{train}}$ and noise vector $\mathbf{z} \sim p(\mathbf{z})$, the critic learns to distinguish the generated data $G_\theta(\mathbf{x}_i, \mathbf{z})$ paired along with $\mathbf{x}_i$ against another pair $(\mathbf{x}_i, \mathbf{x}_j)$. Here, the point $\mathbf{x}_j$ is chosen such that the points $\mathbf{x}_i$ and $\mathbf{x}_j$ have the same label in $D_{\mathrm{cl}}$, i.e., $y_i = y_j$. Hence, the critic learns to classify pairs of (real, real) and (real, generated) points while encouraging the generated points to be of the same class as the point being conditioned on. For the generated data, the label $y$ is assumed to be the same as the class of the point that was used for generating the data. We refer the reader to Antoniou et al. (2017) for further details.

Given a DAGAN model, we additionally require training a binary classifier for estimating importance weights and a multi-class classifier for subsequent classification. The architecture for both these use cases follows prior work in meta learning on Omniglot (Vinyals et al., 2016). Except for the final output layer, the architecture consists of 4 blocks of 3x3 convolutions and 64 filters, followed by batch normalization (Szegedy et al., 2016), a ReLU non-linearity and 2x2 max pooling. Learning was done for 100 epochs using the Adam optimizer with default parameters and a learning rate of 0.001 with a batch size of 32.

### A.3 MODEL-BASED OFF-POLICY POLICY EVALUATION

We evaluate over three envi(onments, including HalfCheetah, Swimmer and HumanoidStandup (Figure 4. Both HalfCheetah and Swimmer rewards the agent for gaining higher horizontal velocity; HumanoidStandup rewards the agent for gaining more height via standing up. In all three environments, the initial state distributions are obtained via adding small random perturbation around a certain state. The dimensions for state and action spaces are shown in Table 4.

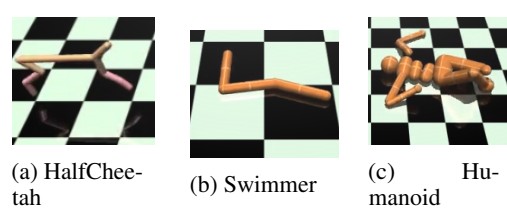

(a) HalfChee-
tah
(b) Swimmer
(c)      Hu-
manoid

Figure 4: Environments in OPE experiments.

Our policy network has two fully connected layers with 64 neurons and tanh activations for each layer, where as our transition model / classifier has three layers of 500 neurons with swish activations.

Table 4: Statistics for the environments.

| Environment | # State dim. | # Action dim |
|---|---|---|
| HalfCheetah | 17 | 6 |
| HumanoidStandup | 376 | 17 |
| Swimmer | 8 | 2 |

We obtain our evaluation policy by training with PPO for 1M timesteps, and our behavior policy by training with PPO for 500k timesteps. Then we train the dynamics model $P_\theta$ for 100k iterations with a batch size of 128. Our classifier is trained for 10k iterations with a batch size of 250, where we concatenate $(s_{t+1}, s_t, a_t)$ into a single vector. We also experimented with other hyperparameters in reasonable regions and the results do not vary significantly.

