# OpenReview forum: "Bias Correction of Learned Generative Models via Likelihood-free Importance Weighting"
_ICLR.cc/2019/Workshop/DeepGenStruct — DeepGenStruct 2019_

### Official Review · AnonReviewer2 · 2019-04-06
**Good idea, good experiments**

**Rating:** 4
**Confidence:** 3

**Review:**

Pros:
  * shows improvements in data augmentation, choosing samples from generative models, and model based policy elevation.
  * well written.
  * experiments in 3 domains, for evaluation, data augmentation and policy evaluation.

Cons:
  * missing ablation experiments for what made the density ratio trick work: self normalization, architecture, etc.
  * for policy evaluation, missing baseline with model free evaluation - by storing the log probs / policy that was used to obtain a particular trajectory, as standard in model free off policy learning.
  * for the generative model evaluation experiments, the starting point is a pretrained classifier. Would be good to know what happens when a classifier is trained from scratch.
  * lacking a bigger discussion for what happens when the two distributions lack common support.

Missing citations:
Azadi S, Olsson C, Darrell T, Goodfellow I, Odena A. Discriminator rejection sampling. arXiv preprint arXiv:1810.06758. 2018 Oct 16.  - a discussion of using the discriminator in GANs
Rosca M, Lakshminarayanan B, Mohamed S. Distribution matching in variational inference. arXiv preprint arXiv:1802.06847. 2018 Feb 19. - experimental work showing the failure modes of the density ratio trick.

---

### Official Review · AnonReviewer1 · 2019-04-18
**Interesting paper, but the experiments could be improved**

**Rating:** 3
**Confidence:** 3

**Review:**

The paper proposes to use importance sampling to debias expectations computed using deep generative models. They demonstrate the usefulness of the idea on tasks such as goodness-of-fit testing, data augmentation and model-based off-policy evaluation. While the underlying ideas have been proposed earlier, I think the combination of ideas (estimating density ratio using a deep neural network and using that to debias deep generative models) is novel and interesting.

Major comments:

- The papers by Azadi et al. 2018 and Turner et al. 2018 propose rejection sampling for GANs. Given the similarity between rejection sampling and importance sampling (the latter is a soft-weighted version of the former), I wish the authors had more prominently discussed the connections between their paper and these papers, and empirically compared to rejection sampling in some of their experiments. I also feel noise contrastive estimation deserves a more prominent discussion.

- “Synthetic experiment” in page 5: how is the uncertainty computed?

- Page 6: “ensured that the classifiers used were well-calibrated” how?

- The experiments are not very compelling and could be improved. Table 2: Why is D_g + IW so poor?

- Importance sampling is known to suffer from high variance. How do you address this?

Minor comments:

- Typo in page 5: “parameteric”

---

### Decision · Program_Chairs · 2019-04-19
**Acceptance Decision**

Accept